# How to Form a Successful Team for the Novel Olympic Triathlon Discipline: The Mixed-Team-Relay

**DOI:** 10.3390/jfmk7020046

**Published:** 2022-06-02

**Authors:** Claudio Quagliarotti, Daniele Gaiola, Luca Bianchini, Veronica Vleck, Maria Francesca Piacentini

**Affiliations:** 1Department of Movement, Human and Health Sciences, University of Rome ‘Foro Italico’, 00135 Rome, Italy; c.quagliarotti@studenti.uniroma4.it (C.Q.); dani.gaiola@gmail.com (D.G.); luca.bianchini.tri@gmail.com (L.B.); 2Centre for the Interdisciplinary Study of Human Performance (CIPER), Faculdade de Motricidade Humana, University of Lisbon, 1499-002 Lisbon, Portugal; vvleck@fmh.ulisboa.pt

**Keywords:** elite triathletes, practical suggestions, coaches, predict performance, super-sprint

## Abstract

The triathlon Mixed-Team-Relay (MTR) is a new race format present for the first time at the Tokyo Olympic Games in 2021. The results of the ITU Triathlon Mixed Relay World Championship from 2014 to 2019 were collected to provide practical suggestions for forming a successful MTR, such as the importance of each leg and discipline on MTR and Super-Sprint performance. The total relay time (T_relay_), the time of each team member (leg-from 1 to 4) (T_leg_), and the time of each single discipline (swim, T1, cycle, T2, run) were collected from the official website. Inferential analysis was performed to assess prediction and differences between variables. Leg 3 was shown to be the most important to predict T_relay_ (0.41), which is also the slower. For both T_relay_ and T_leg_, cycling resulted as the most important (>0.60) and longer (~52%) portion, followed by running and swimming. However, higher importance in swimming was found in successful teams compared to running. For a successful MTR, we suggest: (a) use short-distance specialized triathletes; (b) strengthen cycling and swimming; (c) position in legs 1 and 2 athletes capable of racing in a group; in legs 3 and 4 athletes capable of racing in a non-drafting situation.

## 1. Introduction 

A triathlon is defined as a multisport that consists of consecutive swim, cycle, and run disciplines and two transition phases (T1: swim to cycle transition, T2: cycle to run transition) [1]. The ability to optimally link the three disciplines has been described as an important determinant of success [2]. A triathlon is raced over a variety of distances and formats, including the draft legal (1.5-km swimming, 40-km cycling, and 10-km running) Olympic distance event, featured in every Olympic Games since Sydney 2000. In 2009 the International Triathlon Union (now World Triathlon) introduced the Mixed-Team-Relay (MTR), in which each of four triathletes (in the order female–male–female–male) has to complete a super-sprint triathlon (involving 250–300 m swimming, 5–8 km cycling and 1.5–2 km running) [3]. This relay was already present at the Youth Olympic Games from the first edition in 2010, together with the sprint distance (individual race), and has been included for the first time in the Summer Olympics program at Tokyo 2020. The highest level of MTR racing besides the Olympic Games is the annual ITU Triathlon Mixed Relay World Championship, hosted in Hamburg (Germany) since 2013. The first seven teams in the World MTR Olympic Qualification Ranking are also eligible to participate in the following Olympic Games.

In an Olympic distance competition, the successful top level triathletes aimed to exit in the first pack after swimming [4,5], to maintain their position in the pack during the cycling [6] and to perform a maximal effort in the running section to determine the final results [7]. Recently, it was shown how the importance of each discipline for overall triathlon performance is different based on the triathlon distance [8]. In particular, cycling resulted as the higher predictor in sprint distance, swimming in Olympic and running in the full long distance, although the proportion of time spent swimming (10–17%), cycling (54–56%) and running (30–40%) remains similar between the triathlon distances [7,8]. To the best of our knowledge, very few studies to date have investigated the super-sprint triathlon distance [9,10] and no studies on MTR have been published. At the time of writing, there appears to be only one reference to unpublished pacing-related data for one team in the literature [11].

Therefore, this study aimed to provide practical suggestions for forming a successful MTR based on the results of the lasts World Championships. In particular, we evaluated the role of each leg and portion (disciplines and transitions) on the overall MTR and on the single-leg (super-sprint), hypothesizing a higher importance of cycling, followed by running and swimming.

## 2. Materials and Methods

### 2.1. Data Collection

The results of the ITU Triathlon Mixed Relay World Championship from 2014 to 2019 were collected (triathlon.org/results) [12]. All the championships were hosted in Hamburg (Germany) on the same course (300 m swimming, 6.6 km cycling and 1.7 km running). In particular, the time of overall MTR (T_relay_) of each complete leg (from 1 to 4) (T_leg_) and of each discipline (T_discipline_) and transition (T_transition_) (swim, T1, cycle, T2, run) were collected. Teams that dropped out of the competition or were disqualified were not included in the analysis. All the study data were in the public domain; therefore informed consent was not required [12].

### 2.2. Data Analysis

We included in the analysis a new transition zone that was called a touch-zone, which starts when the finishing triathlete “touches” his/her team member, and ends when the second triathlete enters into the water for the beginning of his/her swim discipline. Unfortunately, the touch-zone time was not provided by the official race results. The mean touch-zone time of each edition for each leg was estimated by video-analysis. Two independent operators uploaded the video recording of the races [13] on Kinovea v.0.8.15 (Charmant and Contrib., Bordeaux, France) and analysis was performed using frame-by-frame playback; this method was previously validated with the error of technical measurement within 0.02 s [14,15]. The touch-zone mean time in legs 2–4 for each Championship edition was estimated (Appendix A) and the relative time value was subtracted from the completed leg and from the swim time of all triathletes competing in legs 2–4.

The percentage values of each complete leg time (from 1 to 4) relative to total relay time and of each discipline and transition (swim, T1, cycle, T2, and run) relative to the complete leg time were calculated.

The team relay sample was divided into: all teams (total) and teams finishing in the first three positions (G1, medalists).

### 2.3. Statistical Analysis

All statistical analyses were performed using SPSS v.25.0 (IBM, Chicago, IL, USA) for Windows and the probability of statistical significance was set as *p* ≤ 0.05. The homogeneity of the data were assessed by the Shapiro–Wilk test.

Kruskal–Wallis H-test was performed for identifying the statistical difference between three or more groups (i.e., percentage of T_relay_, percentage of T_leg_) and when a significant difference was found, the Mann–Whitney U-test was the post-hoc test that was utilized for pairwise comparisons. Man–Whitney U-test was performed for identifying the statistical difference between two groups (i.e., T_leg2_ vs. T_leg4_ (males) and T_leg1_ vs. T_leg3_ (females). The effect size was evaluated by eta partial square (*η^2^*) for Kruskal–Wallis H-test and by the biserial coefficient (*r*) for the Mann–Whitney U-test. The value of *η^2^* and *r* were considered as: small (0.100–0.299), moderate (0.300–0.499), large (0.500–0.699), very large (0.700–0.899) and extremely large (≥0.900) [16,17].

Automatic regression linear models were performed for all teams and G1 group to determine the importance of each T_discipline_ in the prediction of T_relay_ and T_leg_.

All data were presented as median ± interquartile range.

## 3. Results

Data from 92 teams were collected, for a total of 368 triathletes (184 male and 184 female). A total of 18 teams were inserted into G1. Figure 1 presents the leg analysis on T_relay_, the analysis of disciplines and transitions portion is presented in Figure 2 and Figure 3. All of the regression models showed a very high predictive accuracy value (from 95.0% to 100%) for T_relay_ or T_leg_.

Detailed data are provided in the Appendix A (Appendix A).

## 4. Discussion

The objective of the present study was to provide practical suggestions for forming a successful MTR by analyzing the results of the latest World Championships.

The main finding of the study was that the percentage of T_leg_ was significantly different between all legs (Figure 1A). Interestingly, higher T_leg_ were shown for the last two relay members (leg 3 for females and leg 4 for males), indicating a lower performance (Figure 1B). Remarkably, despite the last two legs showing lower performances, they demonstrated higher importance for the T_relay_ prediction (Figure 1C), in particular leg 3 (0.41), which is also the slower leg of the relay. Indeed, paraphrasing Thomas Reid’s comment that “the chain is only as strong as its weakest link” [18], it seems that the good results of a relay depend on the level of the slower team members.

These findings need to be interpreted with caution. The higher performance of the first triathletes might be explained by the interpretation of the MTR race strategy. As already reported for the Olympic distance race, the position of the triathlete during the first part of the race appears to be critical for the development of the race and the finish result [5]. Stronger triathletes in the first legs guarantee a sufficient gap from the other teams that can be maintained by the following teammates. Meanwhile, it is reasonable to affirm that in the second half of the competition triathletes race more in a non-drafting modality because of the gaps that have been previously created, reducing the possibility to form groups or packs. For this reason, the time to complete the race could be higher in legs 3 and 4, because of a reduced possibility to benefit from drafting and/or changing position regularly in cycling. It is well known that drafting during a triathlon (swim and cycle) decreases energy cost and improves overall performance [6,19,20,21].

Therefore, it seems wiser to position in legs 1 and 2 those triathletes that are stronger while racing in a group and capable of taking advantage from drafting. The last two team members should be strong triathletes in non-draft legal conditions and good pacers, both while swimming and while cycling.

Cycling resulted in being both the longer time portion of the race (~52%) (Figure 3A) and was the most important predictor of T_relay_ (0.66) and T_leg_ (0.64) (Figure 2). The predominant importance of cycling may be explained by the time spent by triathletes in the cycle portion, which is more than double compared to the other disciplines (~18% swim, ~3% T1, ~51–52% cycle, ~1.8% T2, and ~25% run) (Figure 3A,B). These percentages are comparable to what has already been reported for the Olympic distance (15% swim, 55% cycle, and 30% run) [7,8].

For the T_leg_, following cycling (0.64), similar importance was found in the running (0.17) and swimming (0,16) portions (Figure 2B). The same importance trend was found in the sprint distance, where the cycling portion resulted in being the best predictor (>0.70) followed by running (>0.1) and swimming (<0.1), while swimming has been reported to be the best predictor for the Olympic distance in elite triathletes [8]. Indeed, it seems that swimming has an increased role in the super-sprint performance, compared to the sprint distance. This was corroborated also by the slightly higher importance of the swimming portion (0.16) in G1 compared to running (0.12) (Figure 2B). These results support the idea that a fast swimming portion may become increasingly important in elite athletes competing in an MTR [22].

If we compare the legs of the same sex (i.e., 1–3 and 2–4) the results highlight that for the males there was a difference in performance between all three disciplines, but for the women, this was only the case during cycling (Figure 3C,E). The results are surprising and further studies are recommended.

The total race time of an MTR is ~80 min, with a total T_leg_ of ~19 min for elite males and ~21 for elite females (Figure 1B), while the duration of an Olympic distance event varies between 106–110 min for elite males and between 119–121 for elite females [6]. The physical demands on the triathlete substantially differ based on race length (i.e., super-sprint and Olympic), where short triathlon courses mean a higher racing intensity that exacerbates particularly the central and peripheral mechanisms responsible for fatigability, compared to a longer distance [9,10]. Specifically, repetitive high-intensity efforts due to the technical courses are necessary in shorter distances [21], both with a higher technical level, such as transitions and cycle portions [10], compared to longer distances requiring steady-state paced efforts [21]. For this reason, the specific skills necessary to excel in the shorter super-sprint distance are quite different from those of the Olympic distance. However, to date almost all triathletes included in an MTR race are Olympic distance specialized. As seen in the last XXXII Olympic Games, 60 out of 64 triathletes competing in the MTR had also participated in the Olympic distance competition.

These results suggest that the MTR is not a single “entity” performance similar to an Olympic distance, but rather a sum of four super-sprint races. Moreover, the use of short-distance specialized triathletes seems to be recommended when selecting team members. Although all disciplines are important to T_leg_, cycling and swimming resulted in being the most important predictors of performance.

The present study investigated the official competition results. Therefore, future experimental studies focused on super-sprint triathlon distance and/or MTR are recommended. Moreover, although there are limitations in identifying correctly the real-time performance of each leg for the missing data-time of the so-called “touch-zone”, we believe our work could be a springboard for future studies on this novel Olympic discipline. Finally, it is important to note that starting from January 2022 to December 2024 the starting order of the MRT has changed (males in the first-third legs and females in the second-fourth legs (rule 16.2) [1]). Despite this change, the triathletes’ characteristics should remain as suggested in the present study (i.e., short distance specialized, first legs skilled in racing in a group, latest legs able to pace in a non-drafting situation), however the importance of each T_leg_ predictor may be impacted. For this reason, future studies should focus on this new format.

## 5. Conclusions

The MTR is a candidate for one of the most spectacular disciplines in triathlon. However, poor knowledge is available about this novel event, which presumably will gain increasing interest after its debut at the Olympic Games in Tokyo 2020.

The use of short-distance specialized triathletes seems to be recommended while selecting team members. In particular, the choice of triathletes skilled in racing in a group seems to be better positioned in legs 1 and 2, whilst triathletes able to pace correctly and to race in a non-drafting situation seem to be better positioned in legs 3 and 4

For training purposes, the cycling and swimming portions seem to be the most useful trainable disciplines to obtain the best overall MTR performance, but also for the single super-sprint distance.

## 6. Practical Applications

As practical recommendations for coaches to form a successful team for the novel triathlon mixed-relay, we suggest the following:Use short-distance specialized triathletes;Strengthen cycling and swimming as they seem to be the major predictors of success;Position athletes as follows: in legs 1 and 2, athletes skilled in racing in a group; in legs 3 and 4, athletes able to pace correctly and to race in a non-drafting situation.

## Figures and Tables

**Figure 1 jfmk-07-00046-f001:**
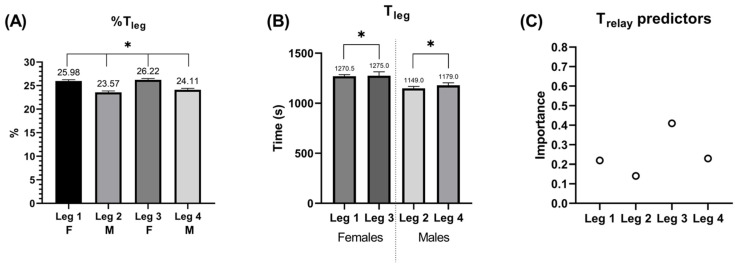
(**A**) Leg time percentage on overall relay performance, (**B**) Total time comparisons between same-sex legs, (**C**) Importance of each leg to predict overall relay performance.* Significantly different (Median ± interquartile range; *p* ≤ 0.05).

**Figure 2 jfmk-07-00046-f002:**
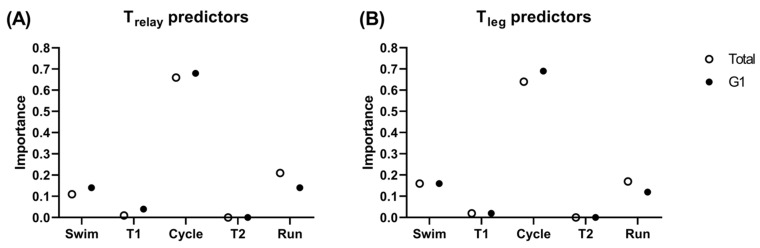
Importance of each discipline and transition to predict (**A**) overall relay time and (**B**) leg time of all teams (total) and G1 (medalists).

**Figure 3 jfmk-07-00046-f003:**
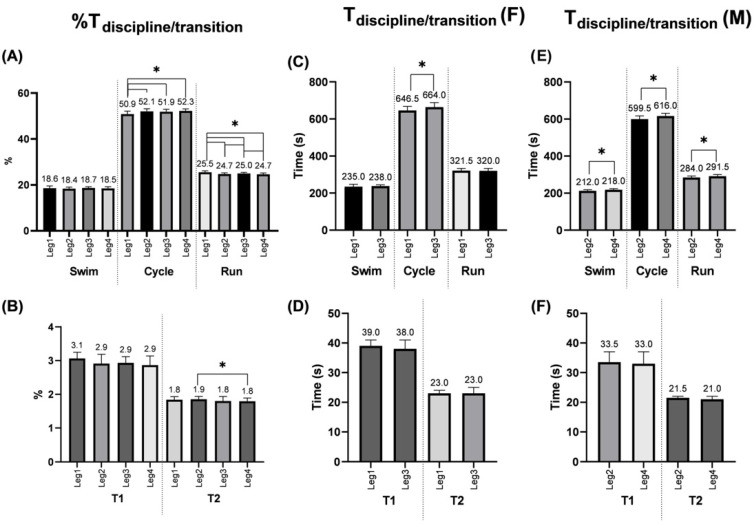
Percentage of overall relay time (**A**,**B**) and time for each leg ((**C**,**D**) females, (**E**,**F**) males) for each discipline and transition. * Significantly different (Median ± interquartile range; *p* ≤ 0.05).

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
