# Peer review of "How to Form a Successful Team for the Novel Olympic Triathlon Discipline: The Mixed-Team-Relay"

_jfmk, 2022, doi:10.3390/jfmk7020046_

Round 1

Reviewer 1 Report

The authors collected the results of the ITU Triathlon Mixed Relay World Championship from 2014 to 2019 in order to provide practical suggestions for forming a successful MTR. Based on their findings, they suggested to „a) use short-distance specialized triathletes; b) focus training more on cycling and swimming; c) position in legs 1 and 2 athletes capable of racing in group; in legs 3 and 4 athletes capable to race in a non-drafting situation“ for a successful MTR.

These findings are in my opinion of relevance to sports kinesiology and would fit the scope of the journal. There are, however, minor concerns which should be addressed.

Please, clearly formulate the aim of the study:

L11-12: To provide practical suggestions for forming a successful MTR, the results of the ITU Triathlon Mixed Relay World Championship from 2014 to 2019 were collected.

Consider to reformulate this sentence:

L18: However, higher importance in swimming was found in successful teams.

The introduction as it stands does not concisely focus on aspects that would argue in favor of the novelty (from a scientific point of view) and the high relevance of this study. It should be considerably restructured. General information about triathlon should be reduced or removed. Instead, analysis of the literature related to swimming, cycling and running performance in triathlon should be provided.

L25-41: Triathlon is defined as a multisport that consists of a consecutive swim, cycle, and run disciplines and two transition phases (T1: swim to cycle transition, T2: cycle to run transition). The ability to optimally link the three disciplines has been described as an important determinant of success [1]. Triathlon is raced over a variety of distances and formats, including the draft legal (1.5-km swimming, 40-km cycling, and 10-km running) Olympic distance event, featured in every Olympic Games since Sydney 2000. In 2009 the International Triathlon Union (now World Triathlon) introduced the Mixed-Team-Relay (MTR), in which each of four triathletes (in the order female-male-female-male) has to complete a Super-Sprint triathlon (involving 250-300m swimming, 5-8km cycling and 1.5-2km running) [2].

This relay was already present at the Youth Olympic Games from the first edition in 2010, together with the Sprint distance (individual race), and has been included for the first time in the Summer Olympics program at Tokyo 2020.

The highest level of MTR racing besides the Olympic Games is the annual ITU Triathlon Mixed Relay World Championship, hosted by Hamburg (Germany) since 2013. The first 7 teams in the World MTR Olympic Qualification Ranking are also eligible to participate in the following Olympic Games.

I would suggest to reformulate the aim of the study.

L46-50: Therefore, this study aimed to provide practical suggestions for forming a successful MTR by (in order of specificity): 1) identifying the importance of each leg on overall MTR performance, 2) evaluating the role of each portion (disciplines and transitions) on the overall MTR and 3) on single-leg (Super-Sprint) performances. Analysis will be done comparing total results with successful teams.

I would suggest to include this part into the „Discussion“.

L190-195: Starting from January 2022 the starting order of the MRT (males in the first-third legs and females in the second-fourth legs (rule 16.2)[20]) has changed. Despite this change, the triathletes' characteristics should remain as suggested in the present study (i.e. shorth distance specialized, first legs skilled in racing in a group, latest legs able to pace in a non-drafting situation) however the importance of each Tleg predictor, may be impacted. For this reason, future studies should focus on this new format.

Relevant limitations of this research should also be discussed.

Author Response

Reviewer 1

Comments and Suggestions for Authors

The authors collected the results of the ITU Triathlon Mixed Relay World Championship from 2014 to 2019 in order to provide practical suggestions for forming a successful MTR. Based on their findings, they suggested to „a) use short-distance specialized triathletes; b) focus training more on cycling and swimming; c) position in legs 1 and 2 athletes capable of racing in group; in legs 3 and 4 athletes capable to race in a non-drafting situation“ for a successful MTR.

These findings are in my opinion of relevance to sports kinesiology and would fit the scope of the journal. There are, however, minor concerns which should be addressed. 
A: We thank the reviewer for the  comments and suggestions, and the appreciation for the present manuscript. We hope to improve the relevance of the manuscript based on the reported suggestions.

Please, clearly formulate the aim of the study:

L11-12: To provide practical suggestions for forming a successful MTR, the results of the ITU Triathlon Mixed Relay World Championship from 2014 to 2019 were collected.

A: We now rephrased this sentence and made the aim of the study clearer.

Consider to reformulate this sentence:

L18: However, higher importance in swimming was found in successful teams.

A: We rephrased the sentence.

The introduction as it stands does not concisely focus on aspects that would argue in favor of the novelty (from a scientific point of view) and the high relevance of this study. It should be considerably restructured. General information about triathlon should be reduced or removed. Instead, analysis of the literature related to swimming, cycling and running performance in triathlon should be provided.

L25-41: Triathlon is defined as a multisport that consists of a consecutive swim, cycle, and run disciplines and two transition phases (T1: swim to cycle transition, T2: cycle to run transition). The ability to optimally link the three disciplines has been described as an important determinant of success [1]. Triathlon is raced over a variety of distances and formats, including the draft legal (1.5-km swimming, 40-km cycling, and 10-km running) Olympic distance event, featured in every Olympic Games since Sydney 2000. In 2009 the International Triathlon Union (now World Triathlon) introduced the Mixed-Team-Relay (MTR), in which each of four triathletes (in the order female-male-female-male) has to complete a Super-Sprint triathlon (involving 250-300m swimming, 5-8km cycling and 1.5-2km running) [2].

This relay was already present at the Youth Olympic Games from the first edition in 2010, together with the Sprint distance (individual race), and has been included for the first time in the Summer Olympics program at Tokyo 2020.

The highest level of MTR racing besides the Olympic Games is the annual ITU Triathlon Mixed Relay World Championship, hosted by Hamburg (Germany) since 2013. The first 7 teams in the World MTR Olympic Qualification Ranking are also eligible to participate in the following Olympic Games.

 A: We now provide further information about the analysis of the literature related to each discipline. We decided to maintain the literature analysis of triathlon because,  from our point of view, it helps understand the aim of our study

I would suggest to reformulate the aim of the study.

L46-50: Therefore, this study aimed to provide practical suggestions for forming a successful MTR by (in order of specificity): 1) identifying the importance of each leg on overall MTR performance, 2) evaluating the role of each portion (disciplines and transitions) on the overall MTR and 3) on single-leg (Super-Sprint) performances. Analysis will be done comparing total results with successful teams.

 A: We have now rephrased this part of the text.

I would suggest to include this part into the „Discussion“.

L190-195: Starting from January 2022 the starting order of the MRT (males in the first-third legs and females in the second-fourth legs (rule 16.2)[20]) has changed. Despite this change, the triathletes’ characteristics should remain as suggested in the present study (i.e. shorth distance specialized, first legs skilled in racing in a group, latest legs able to pace in a non-drafting situation) however the importance of each Tleg predictor, may be impacted. For this reason, future studies should focus on this new format.

 A: We have moved this paragraph to the discussion.

Relevant limitations of this research should also be discussed.

A: We have added a paragraph on the limitations of the study at the end of the manuscript.

Reviewer 2 Report

Dear authors,

This study aimed to provide practical suggestions for forming a successful MTR by (in order of specificity): 1) identifying the importance of each leg on overall MTR performance, 2) evaluating the role of each portion (disciplines and transitions) on the overall MTR and 3) on single-leg (Super-Sprint) performances. There are some specific changes and suggestions that should be made to improve the quality of the paper.

Best regards.

Author Response

Reviewer 2

This study aimed to provide practical suggestions for forming a successful MTR by (in order of specificity): 1) identifying the importance of each leg on overall MTR performance, 2) evaluating the role of each portion (disciplines and transitions) on the overall MTR and 3) on single-leg (Super-Sprint) performances. There are some specific changes and suggestions that should be made to improve the quality of the paper.

A: We thank the reviewer for the  comments and suggestions, and the appreciation for the present manuscript. We hope to improve the relevance of the manuscript based on the reported suggestions.

GENERAL QUESTIONS

  • -  Pg01Ln25 – insert a reference in this sentence “Triathlon is defined as a multisport that consists of a consecutive swim, cycle, and run disciplines and two transition phases (T1: swim to cycle transition, T2: cycle to run transition)”.

A: We now provided a reference

  • -  The authors of the introduction section should develop a little more on the subject, that is, talk a little about the studies that exist and what results they reached in these studies.

A: We thank the reviewer for the suggestion and developed the introduction as suggested by both reviewers. .

  • -  Insert in the introduction section, what are the hypotheses of the study.

A: We have now included the hypotheses at the end of the introduction section.

  • -  Authors should create a new section with the title “participants”. The new section must be before the data collection.

A: We decided not to include this specific section. The analysis was performed retrieving data from free online data and the number of teams or athletes is a result of the method that has been utilized to retrieve this information

-  Insert in the participants' section the approval of the ethics committee where the study was carried out.

A: The study is based on free available published online data. No ethics committee approval was necessary to conduct the study, as reported in Pg02Ln65-66. As done in other published articles as: Baldassarre et al. 2019 doi: https://doi.org/10.3390/jfmk4010015 ; Boccia et al. 2021 doi: https://doi.org/10.1123/ijspp.2020-0090 ; Filipas et al. 2018 doi: https://doi.org/10.1519/JSC.0000000000002873 ; Piacentini et al. 2019 doi: https://doi.org/10.3390/sports7040076 )

  • -  Pg02Ln67 - Enter the country and location of the software.

A: this information has been added

  • -  Pg02Ln70 - What is table S1.1...it is not in the manuscript.

A: The table S1 and later are inserted as Supplementary Materials, and can be found on the journals website.

Pg02Ln80 - The “p” must be in italics.

A: We changed accordingly in the whole manuscript and Supplementary Materials

  • -  Describe in the results section, a little more about the results obtained, that is, insert the values of significant results.

A: We inserted all the statistical results in the Supplementary Materials in order not to overload the reader while  reading the main text

  • -  This information “Data from 92 teams were collected, for a total of 368 triathletes (184 male and 184 female). A total of 18 teams were inserted into G1” should be in the new “participants” section.

A: We prefer to leave this information in the results section. Indeed, the number of teams/athletes collected is a result of the methods applied and not a value stated before the data collection/analysis.

  • -  Insert in the legend of figures 1 and 3, which level of significance was used.

A: We have included this information

  • -  The first paragraph of the discussion section should be a reminder of the purpose of the study. Please enter this information.

A: We have now included as a first sentence the aim of the study 

-  Pg04Ln117 – insert a reference in this sentence “Indeed, paraphrasing Thomas Reid ́s comment that “the chain is only as strong as its weakest link”, it seems that the good results of a relay depend on the level of the slower team members”.

A: We inserted a reference

-  Insert at the end of the discussion section what form the limitations of the study.

A: We reported the limitations of the study in the discussion section.

  • -  References should not be included in the conclusion. Please correct.

A: We now divided the “conclusion”/”Practical applications” and “reference” sections

Round 2

Reviewer 2 Report

Dear authors,

Congratulations on your choice of topic. The manuscript has significantly improved since the last revision.

Best regards